# The Impact of Semicarbazide Sensitive Amine Oxidase Activity on Rat Aortic Vascular Smooth Muscle Cells

**DOI:** 10.3390/ijms24054946

**Published:** 2023-03-03

**Authors:** Vesna Manasieva, Shori Thakur, Lisa A. Lione, Anwar R. Baydoun, John Skamarauskas

**Affiliations:** 1Department of Metabolism, Digestion and Reproduction, School of Medical Sciences, Imperial College, London SW7 2AZ, UK; 2Department of Clinical, Pharmaceutical and Biological Science, School of Life and Medical Sciences, University of Hertfordshire, Hatfield AL10 9AB, UK; 3Faculty of Health and Life Sciences, School of Pharmacy, De Montford University, Leicester LE1 9BH, UK

**Keywords:** vascular smooth muscle cells, semicarbazide-sensitive amine oxidase, methylamine, aminoacetone, formaldehyde, methylglyoxal, hydrogen peroxide, reactive oxygen species

## Abstract

Semicarbazide-sensitive amine oxidase (SSAO) is both a soluble- and membrane-bound transmembrane protein expressed in the vascular endothelial and in smooth muscle cells. In vascular endothelial cells, SSAO contributes to the development of atherosclerosis by mediating a leukocyte adhesion cascade; however, its contributory role in the development of atherosclerosis in VSMCs has not yet been fully explored. This study investigates SSAO enzymatic activity in VSMCs using methylamine and aminoacetone as model substrates. The study also addresses the mechanism by which SSAO catalytic activity causes vascular damage, and further evaluates the contribution of SSAO in oxidative stress formation in the vascular wall. SSAO demonstrated higher affinity for aminoacetone when compared to methylamine (Km = 12.08 µM vs. 65.35 µM). Aminoacetone- and methylamine-induced VSMCs death at concentrations of 50 & 1000 µM, and their cytotoxic effect, was reversed with 100 µM of the irreversible SSAO inhibitor MDL72527, which completely abolished cell death. Cytotoxic effects were also observed after 24 h of exposure to formaldehyde, methylglyoxal and H_2_O_2_. Enhanced cytotoxicity was detected after the simultaneous addition of formaldehyde and H_2_O_2_, as well as methylglyoxal and H_2_O_2_. The highest ROS production was observed in aminoacetone- and benzylamine-treated cells. MDL72527 abolished ROS in benzylamine-, methylamine- and aminoacetone-treated cells (**** *p* < 0.0001), while βAPN demonstrated inhibitory potential only in benzylamine-treated cells (* *p* < 0.05). Treatment with benzylamine, methylamine and aminoacetone reduced the total GSH levels (**** *p* < 0.0001); the addition of MDL72527 and βAPN failed to reverse this effect. Overall, a cytotoxic consequence of SSAO catalytic activity was observed in cultured VSMCs where SSAO was identified as a key mediator in ROS formation. These findings could potentially associate SSAO activity with the early developing stages of atherosclerosis through oxidative stress formation and vascular damage.

## 1. Introduction

Semicarbazide-sensitive amine oxidase (SSAO) is a copper-rich amine oxidase encoded by the amine oxidase copper containing 3 (*Aoc3*) gene, and it exists as both a soluble- and membrane-bound transmembrane protein, also known as vascular adhesion protein 1 (VAP-1). Soluble SSAO is a result of the proteolytic cleavage of the membrane-bound VAP-1. During this process, anchored molecules are released into the bloodstream by shedding from the membrane through a metalloproteinase-dependent activity [1,2]. SSAO converts primary amines into their corresponding aldehydes while also generating hydrogen peroxide and ammonia. Being a vascular enzyme SSAO is highly expressed in the vascular endothelial and in smooth muscle cells. In endothelial cells, SSAO is localized in the intracellular/cytoplasmic vesicles and its activity in these cells is associated with the development of atherosclerosis, as it induces a leukocyte adhesion cascade into damaged inflammatory sites [3,4]. In smooth muscle cells, SSAO is localized in the caveolae of the plasma membrane. Previously [5], we demonstrated SSAO to be closely associated with another vascular enzyme, lysyl oxidase (LOX), whose alterations in activity and expression have been linked with the early developing stages of atherosclerosis [6,7]. Moreover, we have demonstrated LOX as a regulator of SSAO activity, VAP-1 protein and Aoc3 mRNA expression in early passage rat aortic VSMCs, highlighting SSAO as an important novel therapeutic target for the treatment/prevention of atherosclerosis [5].

Atherosclerosis is defined as a progressive and complex inflammatory disease that develops because of disturbed vascular homeostasis caused by endothelial injury [8]. A lipid profile is an important pathological factor for the development of atherosclerosis. Elevated low-density lipoprotein (LDL) cholesterol and elevated triglyceride-rich lipoproteins (TGRL) and low high-density lipoprotein (HDL) now comprise a major pattern of lipid abnormality in atherosclerosis [9]. Since the development of atherosclerosis is based on metabolic changes in lipid metabolism, major sex-based differences in cholesterol metabolism have been shown to contribute to differences in the pathogenesis of this disease [10]. An SSAO-mediated increase in free radicals provoke an oxidative modification from low-density lipoprotein (LDL) to oxidized low-density lipoprotein (oxLDL) in the vascular wall, which is an important step in the early development stages of atherosclerosis [3,4].

SSAO substrates are aromatic and aliphatic monoamines. They are produced endogenously or absorbed as dietary or xenobiotic substances [11]. Methylamine and aminoacetone are SSAO specific substrates, endogenously produced as short-chain primary amines, and oxidatively deaminated by SSAO to formaldehyde and methylglyoxal. Methylamines have been shown to enhance atherosclerosis in animal models [12] and, in clinical studies, have been associated with cardiovascular risks [13,14,15]. Furthermore, the toxic effects of formaldehyde and methylglyoxal have been widely implicated in cardiovascular pathologies [16,17,18,19]. Both formaldehyde and methylglyoxal are highly reactive aldehydes capable of cross-linking with proteins following a pseudo-first order kinetic [20]. Unlike free radicals, SSAO-derived aldehydes are more stable. This high stability enables methylglyoxal and formaldehyde to diffuse easily and attack intracellular targets that are distant from the point of origin [21].

Early studies have shown that methylamine does not harm endothelial cells at concentrations of up to 100 mmol/L [22]; however, in the presence of SSAO, methylamine has been demonstrated to be cytotoxic due to formaldehyde formation [22]. Formaldehyde can induce cell death by interacting with macromolecular constituents, thus altering cellular structures. Additionally, it has been shown to be a main apoptotic inducer in vascular endothelial cells [23]. The formaldehyde-induced apoptosis in A7r5 cells was detected by chromatin condensation, caspase-3 activation, PARP cleavage and cytochrome c release [23]. In another study, a formaldehyde-driven expression of the pro-apoptotic protein p53 was shown to potentially be an additional mechanism through which formaldehyde induces apoptosis [24]. 

Methylglyoxal is a highly reactive aldehyde. It is also a powerful modifying agent of proteins and DNA and can act as a mediator in the synthesis of advanced glycation end products [25]. It has been previously established that by modifying proteins, as well as forming oxygen free radicals, methylglyoxal can act as a cytotoxic agent and induce apoptosis in cells [25]. Moreover, it contributes to the formation of advanced glycation end product (AGE) by modifying cell proteins non-enzymatically through the Maillard reaction, in which aldehydes and ketones react with ε-amino groups of lysine residues and guanidino groups of arginine residues [25].

H_2_O_2_ is another by-product of an SSAO-catalyzed reaction; if produced above normal physiological levels (1–100 nM), it acts as important contributor to oxidative stress [26]. Recent studies [18,19] have identified H_2_O_2_ as a source of reactive oxygen species (ROS) that can modify low-density lipoprotein (LDL) in the arterial wall and contribute to the development of atherosclerosis. Other studies have addressed H_2_O_2_ as a vasoactive agent with the ability to induce vasoconstriction of resistance vessels and increase vascular tone. Therefore, it can contribute to the development of hypertension [27,28].

Furthermore, an increased H_2_O_2_ production because of enhanced SSAO activity could initiate a signaling cascade which leads to an increased expression of inflammatory cytokines and adhesion molecules in the vascular wall and as such accelerate endothelial damage [29]. H_2_O_2_ is metabolized by catalase and glutathione peroxidase and when produced in large amounts, in the presence of transition metals (particularly iron) it can be converted to toxic hydrogen free hydroxyl radical (^•^OH) via the Fenton reaction (H_2_O_2_ + Fe^2+^ → ^•^OH + OH^–^ + Fe^3+^) [30]. Hydroxyl radicals pose a greater risk comparing to H_2_O_2_ and can directly injure cell membranes and nuclei [31].

SSAO is highly implicated in the pathophysiology of various cardiovascular diseases (CVD), including stroke, myocardial infarction and atherosclerosis, as well as health risks associated with CVD, such as obesity and diabetes [18,19,32,33,34]. Apoptotic VSMCs and increased ROS levels are important hallmarks in the early developing stages of atherosclerosis [9]. Being abundantly present in the vasculature, SSAO is a relatively novel enzymatic discovery associated with cytotoxicity and elevated ROS levels, through production of highly unstable and reactive aldehydes and H_2_O_2_. This study investigates the role of SSAO in the early developing stages of atherosclerosis by exploring its enzymatic activity, cytotoxic effects and contribution in oxidative stress formation in rat aortic VSMCs, using its respective substrates and inhibitors.

## 2. Results 

### 2.1. Active SSAO Induces VSMCs Death 

To assess SSAO activity in rat aortic VSMCs, cells were treated with increasing concentrations of SSAO’s endogenous amines, aminoacetone (AA) and methylamine (M). Figure 1 shows the effect of AA and M on VSMC cell viability after 24 h treatment, and the suppressive effect of SSAO’s irreversible inhibitor MDL72527 on cytotoxicity induced by both amines. Moreover, 0 µM of aminoacetone and methylamine were considered as a vehicle only control. Aminoacetone at 50 µM caused 15% and at 100 µM resulted with 30% cell death comparing to control (Figure 1A). Methylamine at 1000 µM caused 40% cell death comparing to the control (Figure 1B).

### 2.2. Enhanced Cytotoxic Effect Was Observed after Simultaneous Addition of Methylglyoxal and H_2_O_2_, and Formaldehyde and H_2_O_2_

To investigate direct cytotoxic effects from SSAO’s derived products and potential synergism between them, VSMCs were treated with methylglyoxal (MG), formaldehyde (F) and H_2_O_2_. Figure 2 shows the effect of MG, F and H_2_O_2_ on VSMC cell viability after 24 h treatment. MG (50 µM) and F (1000 µM) caused 30–40% cell death comparing to control. H_2_O_2_ at 50 µM caused 30% and at 1000 µM 40% cell death comparing to control (Figure 2A,B). MG and H_2_O_2_ combined caused 70% cell death, with F and H_2_O_2_ causing 60% cell death above control (Figure 2A,B).

### 2.3. SSAO Has Higher Affinity for Aminoacetone Comparing to Methylamine and Converts Both at a Fast Rate in Rat Aortic VSMCs

To understand the level of interaction between SSAO and its endogenous amines, SSAO kinetic parameters were assessed in the presence of aminoacetone and methylamine as substrates. Figure 3 shows SSAO’s reaction rate (nmol H_2_O_2_/h) as function of aminoacetone (A) or methylamine (B) concentration. SSAO demonstrates higher affinity for aminoacetone comparing to methylamine (12.08 µM vs. 65.35 µM), as observed in the Km values, and converts both amines at a fast rate (5 nmol/min for aminoacetone vs. 4 nmol/min for methylamine), as observed in the Vmax values (Figure 3A,B).

### 2.4. SSAO Activity Induces ROS Formation in Rat Aortic VSMCs 

To observe the effect of SSAO on ROS formation in VSMCs and establish an optimal time to measure ROS cells were treated with SSAO substrate amines benzylamine, methylamine and aminoacetone over different time intervals. Further, 5 μM (20 μL/well) DMNQ was used like a positive control. Figure 4A shows gradual increase of ROS production after 5 μM DMNQ treatment over different time intervals. Figure 4B shows the detected ROS production (expressed as percentage of control) versus time (*) and versus different amine treatments (^#^).

ROS levels were measured once again after 30 min of incubation with benzylamine, methylamine and aminoacetone in the presence of SSAO’s irreversible inhibitor MDL72527 and SSAO’s competitive reversible inhibitor βAPN. Figure 5 shows the detected ROS formation (expressed as percentage of control) after 30 min treatment with benzylamine, methylamine and aminoacetone in the presence of MDL72527 or βAPN respectively.

ROS formation was also observed with cell imaging. Figure 6 shows cultured VSMCs stained with ROS red-staining solution to image ROS production. Figure 6A shows ROS production in VSMCs treated with PBS–negative control. Figure 6B shows ROS production in VSMCs treated with DMNQ–positive control. Figure 6C,D shows ROS production after benzylamine treatment (B) and benzylamine in the presence of MDL72527 (B + MDL72527). Figure 6E,F shows ROS production after methylamine treatment (M) and methylamine in the presence of MDL72527 (M + MDL72527). Figure 6G,H shows ROS production after aminoacetone treatment (A), and aminoacetone in the presence of MDL72527 (A + MDL72527). 

### 2.5. SSAO Activity Reduces Total GSH Levels in Rat Aortic VSMCs 

To assess whether SSAO driven ROS production reduces total glutathione (GSH) levels, GSH (nM/mg protein) was detected with a colorimetric recycling assay based on the glutathione recycling system by DTNB (Ellman’s reagent) and glutathione reductase. Figure 7 shows the detected GSH after benzylamine, methylamine and aminoacetone treatment, and after treatment with the amines in the presence of SSAO’s irreversible inhibitor MDL72527 and SSAO’s competitive reversible inhibitor βAPN. Cells in culture medium—without treatment, were considered as control. 

## 3. Discussion

Apoptotic VSMCs and increased ROS levels are distinctive features in the early developing stages of atherosclerotic plaque formation. This study investigates the role of SSAO in the early developing stages of atherosclerosis by exploring its enzymatic activity, cytotoxic effects and contribution in oxidative stress formation in rat aortic VSMCs, using its respective substrates and inhibitors. Our findings show induced VSMCs death after 24 h exposure to 50 & 100 µM aminoacetone, and 1000 µM methylamine, and reversed cytotoxicity after addition of 100 µM irreversible SSAO inhibitor MDL72527, which completely abolished cell death (Figure 1A,B). This suggests that the cytotoxic effects observed here are a consequence of the deamination of methylamine and aminoacetone, a reaction catalyzed by SSAO. A similar effect was observed in previous studies in which another irreversible SSAO inhibitor, MDL-72974A, reversed formaldehyde- [22] and methylglyoxal-induced [35] cell death by inhibiting the deamination of their respective substrates, methylamine and aminoacetone.

Cellular concentrations of methylamine are estimated as <1 mM [36]. Aminoacetone has previously been suggested as cytotoxic at concentrations of ≥100 µM [37]. When the cellular levels of aminoacetone and methylamine reach higher than their physiological range, these amines have been reported to induce cell death in human aortic VSMCs and insulin-producing cells [23,37,38]. SSAO’s driven aminoacetone cytotoxic effect was observed in insulin-producing RINm5f cells where aminoacetone with concentrations 100 and 500 µM reduced cell viability [37]. Our data agrees with these findings; however, it also signifies SSAO driven cytotoxic effect at lower aminoacetone concentrations (50 µM) in rat aortic VSMCs (Figure 1A). This indicates higher VSMCs vulnerability to aminoacetone driven cytotoxic effects, which could be attributed to higher SSAO expression in VSMCs in comparison to pancreatic B cells. In another study, methylamine-induced toxicity was observed at 1 mM in human aortic smooth muscle cells because of SSAO mediated deamination [23]. This was confirmed by observing Caspase-3 activation, PARP cleavage and cytochrome c release [23]. Our data is congruent with this finding by demonstrating a methylamine-driven cytotoxicity at 1 mM in rat aortic VSMCs (Figure 1B). 

Furthermore, our data shows direct cytotoxic effect induced by SSAO generated aldehydes (methylglyoxal and formaldehyde) and H_2_O_2_ (Figure 2A,B). Cytotoxic effects of methylglyoxal and formaldehyde have been previously indicated in endothelial but not in VSMCs [39,40]. Methylglyoxal activity in VSMCs has been associated with the production of advanced glycation end products, such as argpyrimidine [25].

Formaldehyde has been demonstrated as a main inducer of covalent binding between functional groups in lysine residues of protein, and DNA base in rat endothelial cells [39], and methylglyoxal has been shown to induce human umbilical vein endothelial cell death at concentrations of 400–800 µM by downregulating cell cycle associated genes and upregulating the heme-oxygenase 1 (HO-1) [40]. Cellular concentrations for methylglyoxal are between 1–5 µM [21], and for formaldehyde 200–500 µM [41]. Furthermore, physiological range of H_2_O_2_ in the cell is between 1 and 100 nM and high concentrations of 1 and 2 mM can induce cell death by increasing DNA protein crosslinks [39]. Interestingly, it has previously been postulated that the SSAO catalyzed reaction produces equal molar concentrations of cytotoxic aldehyde and H_2_O_2_ and that these by-products act in synergism in inducing cell damage and death [39]. Our study shows cellular toxicity at 50 µM methylglyoxal, 1000 µM formaldehyde, and 50 and 1000 µM H_2_O_2,_ and enhanced cytotoxic effect after simultaneous addition of H_2_O_2_ and aldehydes, which indicates additive rather than synergistic relationship between the same (Figure 2A,B).

After identifying a safe concentration range at which SSAO endogenous amines failed to exert cytotoxic effect on rat aortic VSMCs (Figure 1A,B), we performed additional studies to understand the level of interaction between SSAO and these amines. Our data shows higher SSAO affinity for aminoacetone compared to methylamine, as observed in the Km values, and faster SSAO driven oxidative deamination of aminoacetone compared to methylamine, as observed in the Vmax values (Figure 3A,B). SSAO kinetic parameters have been previously investigated in rat aortic A7r5 cells after addition of methylamine, benzylamine and tyramine as substrates [23]. In contrast to our data, this study demonstrated higher SSAO Vmax (7.32 nmol/min) and smaller SSAO affinity for methylamine (1.04 mM) [23]. The reason for this could be that our study used primary cell line, with a focus on the membrane bound form of the enzyme and not soluble SSAO. Furthermore, our data shows fast generation of methylglyoxal and formaldehyde, because of SSAO catalyzed reaction (Figure 3A,B). The fast generation of methylglyoxal and formaldehyde could damage cell membranes due to auto-oxidation of lipids and fatty acids within the cell [21].

To assess the contribution of SSAO in oxidative stress formation, ROS levels were measured after treatment with SSAO’s respective substrates (benzylamine, methylamine and aminoacetone) and inhibitors (βAPN and MDL72527). This was further correlated with changes in total GSH content. The cytotoxic and ROS formation ability of SSAO derived by-products has been previously highlighted in other studies [42,43,44]. Methylglyoxal was previously shown to increases ROS through AGEs formation [43].

In another study, methylglyoxal-driven ROS was demonstrated as a crucial mechanism for methylglyoxal-induced cytotoxicity in brain endothelial cells, as it suppressed the Akt/hypoxia-inducible factor 1 alpha (HIF-1α) pathway [44]. Interestingly, previous studies have detected synergism between formaldehyde and free radicals in increasing oxidative stress levels and reducing cell viability [42]. Our data shows a significant difference in ROS production after 15-, 30-, 60- and 120-min incubation with the amines, in comparison to the control DMNQ (Figure 4B). Since ROS are defined as relatively short-lived molecules, 30 min was selected as an optimal time to measure ROS production. This is because the detected ROS after methylamine treatment at 30 min was higher in comparison to 15 min, and there was not a significant difference in benzylamine and aminoacetone driven ROS production between 15 and 30 min (Figure 4B). 

Our data shows the highest ROS production after aminoacetone treatment (45 μM), followed by benzylamine (500 μM) and then methylamine (500 μM) (Figure 4B and Figure 5). Aminoacetone is catalytically deaminated to methylglyoxal through SSAO-driven enzymatic reaction. ROS formation has previously been associated with methylglyoxal in vascular endothelial cells [44] and pancreatic beta cells [45]. In vascular endothelial cells, methylglyoxal treatment was shown to increase mitochondrial and total cellular ROS formation [44]. In pancreatic beta cells, methylglyoxal treatment was shown to increase mitochondrial ROS and stimulate overproduction of advanced glycation end products (AGEs) [45]. Since methylglyoxal is an aldehyde produced through SSAO catalyzed reaction in which aminoacetone is oxidatively deaminated to aldehyde (methylglyoxal), our findings correlate with these studies and associate SSAO activity with mitochondrial ROS production in VSMCs.

Benzylamine is also a substrate for lysyl oxidase (LOX), another amine oxidase abundantly present in the VSMCs. While MDL72527 is a specific suicide inhibitor for SSAO, βAPN is a suicide inhibitor for LOX [5], and a competitive reversible inhibitor for SSAO. This explains why a significant reduction in ROS was detected between benzylamine, benzylamine- and βAPN-treated cells (Figure 5). However, the benzylamine-driven ROS reduction after βAPN treatment was smaller (30%) in comparison to the inhibition induced by MDL72527 (80%). Therefore, this data suggests that the ROS detected here is predominantly SSAO driven.

Furthermore, our data shows no significant reduction in ROS in methylamine- and aminoacetone-treated cells after βAPN treatment, and a significant reduction in ROS in methylamine-treated cells (50%) and aminoacetone-treated cells (90%) after MDL72527 treatment (Figure 5). Additionally, the comparison between the two different inhibitor treatments in reducing ROS distinguished MDL72527 as more potent ROS reducing agent in comparison to βAPN (Figure 5), which once again prioritize SSAO over other vascular enzymes, such as LOX in ROS formation. Furthermore, Figure 6 confirms the potency of MDL72527 in inhibiting ROS production in benzylamine-, methylamine- and aminoacetone-treated cells.

ROS levels were correlated with total GSH production (nM/mg protein) in VSMCs previously treated with benzylamine, methylamine, aminoacetone and the substrate amines in the presence of MDL72527 and βAPN. This is because GSH is the main antioxidant that reduces hydrogen peroxide through glutathione peroxidase (GPx) catalyzed reactions [46]. Exposure to ROS could reduce total GSH through its oxidation during which levels of oxidized GSH (GSSG) are increased as a defense mechanism of the cells to counteract ROS [47]. Moreover, hydroxyl radicals could lead to direct oxidation of GSH and consequently GSSG formation. Our data shows significant reduction in total GSH after benzylamine, methylamine and aminoacetone treatment (Figure 7). These findings complement the data from Figure 5, where significant increase in ROS was observed after aminoacetone, benzylamine and methylamine treatment. GSH is an important intracellular antioxidant and, thus, reduction in its levels are paralleled with the generation of different ROS including hydroxyl radicals, superoxide anions, hydrogen peroxide and lipid peroxide [48]. Since the amines used here are specific SSAO substrates (apart from benzylamine that is also deaminated by LOX), and hydrogen peroxide is a by-product of SSAO catalyzed reaction, our study suggests that active SSAO contributes to reduced GSH in rat aortic VSMCs, because of ROS formation.

In contrast to Figure 5, where MDL72527 significantly reduced ROS and its inhibitory potential over ROS was more potent than βAPN, the GSH data does not show significant restoration of total GSH after MDL72527 or βAPN treatment (Figure 7). Previous studies have dissociated the relationship between ROS and GSH by demonstrating that a reduction in GSH is a necessary contributing factor for ROS generation; however, inhibition of ROS by antioxidants does not necessarily restore GSH levels, which indicates independence from the generation of ROS [48].

## 4. Materials and Methods

### 4.1. Reagents

Cell culture reagents were purchased from Fisher Scientific (Loughborough, UK). Unless otherwise stated, chemicals and reagents were purchased from Sigma-Aldrich (Poole, UK).

### 4.2. Animals

The rat model was used due to being closely similar with humans in terms of aortic SSAO activity [49]. Indistinguishable levels of SSAO activity have been previously detected in human and rat arteries (human 2.56 nmol benzaldehyde/min/mg protein *vs.* rat 2.84 nmol benzaldehyde/min/mg protein) [49]. Male Wistar rats (180–220 g) were housed in pairs in standard cages (Tecniplast 2000P) with sawdust (dates and grade 7 substrate) and shredded paper wool bedding with water and food (5LF2 10% protein LabDiet) in the Biological Services Unit at the University of Hertfordshire. The housing environment was maintained at a constant temperature (21 ± 20 °C) and a light-dark cycle (12:12 h). All experiments were carried out in accordance with the University of Hertfordshire animal welfare ethical guidelines and European directive 2010/63/EU and all tissues collected were naïve shared within teaching/research in accordance with the 3Rs.

### 4.3. Cells

The aortic VSMCs were selected due to expressing high levels of SSAO. This study used primary VSMCs because primary cell cultures most closely represent the tissue of origin [50].

### 4.4. Isolation and Characterisation of Rat Aortic VSMCs 

VSMCs were isolated from the rat’s aorta, as per standard protocol, which consists of five steps: isolation of the aortic artery, removal of the fat tissue around the artery, cutting the artery into small tissue blocks, transferring the tissue blocks to cell culture flask and incubation until the cells reach confluency [50].

The rats were euthanized by exposure to carbon dioxide gas in a rising concentration. The aorta was removed and placed in a Dulbecco’s modified eagle medium (DMEM) solution supplemented with 10% Fetal Bovine Serum (FBS (*v*/*v*)), 1% penicillin (100 units mL^−1^), streptomycin (100 μg mL^−1^) and 2 mM L-Glutamine. The aorta was cleaned 3 times with 1× phosphate buffer saline (PBS) and the fat tissue around the artery was removed. The artery was then cut longitudinally, and the intima was softly scrapped to eliminate endothelial cells.

The artery was fixed by pressing it dorsally with a pair of ophthalmic curved tweezers and another pair of ophthalmic curved tweezers was used to separate the media from the artery by pressing and pushing the artery dorsally. The media was then cut into small tissue blocks and transferred to T25 cell culture flask containing Dulbecco’s Modified Eagle’s Medium (DMEM; Gibco^®^, Waltman, MA, USA), supplemented with 10% Fetal Bovine Serum (FBS (*v*/*v*)), 1% penicillin (100 units mL^−1^), streptomycin (100 μg mL^−1^) and 2 mM L-Glutamine. To characterize the cells, the isolated rat VSMCs were stained for the smooth muscle cell marker SM22α, as per standard protocol [51]. Please see Appendix A for isolated and characterized VSMCs images.

### 4.5. Cell Viability Assay

Cell viability was determined with the MTT (3-(4,5-dimethylthiazol-2-yl)-2,5-diphenyltetrazolium bromide) tetrazolium reduction assay, as previously described [52]. Primary rat aortic VSMCs were plated at 5 × 10^4^ in a 96-well plate and allowed to grow for 24–48 h to reach confluence. Confluent cells were pre-treated with different concentrations of aminoacetone or methylamine (dissolved in serum free DMEM), with and without the presence of 100 µM irreversible SSAO inhibitor MDL72527 (Sigma, St. Louis, MO, USA, M2949) for 24 h in a CO_2_ incubator (5% CO_2_ and 95% humidified air) at 37 °C. After incubation with the amines, MTT solution (5 mg/mL) was added to each well, and the plate was incubated for additional 4 h in a CO_2_ incubator (5% CO_2_ and 95% humidified air) at 37 °C. After incubation with MTT the media was removed, and the formazan crystals were dissolved by adding 200 µL isopropanol. The MTT assay was also utilized to determine cell viability after addition of aldehydes and H_2_O_2_. Confluent cells were pre-treated with equal concentrations of methylglyoxal (50 µM), H_2_O_2_, or, methylglyoxal + H_2_O_2_, as well as equal concentrations of formaldehyde (1000 µM), H_2_O_2_ or formaldehyde + H_2_O_2_, (dissolved in serum free DMEM) for 24 h in a CO_2_ incubator (5% CO_2_ and 95% humidified air) at 37 °C before adding MTT solution (5 mg/mL) to each well, followed by further 4 h incubation in a CO_2_ incubator (5% CO_2_ and 95% humidified air) at 37 °C. After incubation with MTT, the media was removed and the formazan crystals were dissolved by adding 200 µL isopropanol. The plates were then wrapped in a foil and placed on a shaker for 15 min. The quantity of formazan was directly proportional to the number of viable cells was measured by recording changes in absorbance at 570 nm, using a spectrophotometric Clario Star^®^ Microplate Reader (BMG Labtech, Ortenberg, Germany).

### 4.6. Amplex Red Assay 

After establishing non-toxic amine concentrations, SSAO kinetic parameters were assessed in the presence of methylamine and aminoacetone as substrates, with the Amplex^®^ red assay previously optimized to detect SSAO activity in this cell type. Rat aortic VSMCs with confluency of ~80–90% were treated with reaction mixture containing 20 μL Amplex^®^ Red, 10 μL horseradish peroxidase (HRPO) and 10 μL clorgyline, supplemented with 0.25 M sodium phosphate buffer at pH 7.4, and different concentrations of methylamine or aminoacetone as substrates. SSAO activity was measured 6 h from the addition of the reaction mixture using excitation 540 nm and emission 590 nm on a Clario Star^®^ Microplate Reader (BMG Labtech). Resorufin was used to measure end-point fluorescence. Next, 2 mM of resorufin stock solution was diluted to a concentration of 1000 μM in a 1× reaction buffer (2 mL of 5× reaction buffer (0.25 M sodium phosphate at pH 7.4), 10 mL distilled water) to yield resorufin standards ranging from 0 to 20 μM. The data was transferred and analyzed on an Excel spreadsheet before preparing a standard curve of resorufin fluorescence (RFU) versus concentration (μM). To express SSAO activity in nmol H_2_O_2_/mL, the fluorescence readings from different time intervals were multiplied by the slope and added by the intercept (both calculated from the linear equation of the resorufin standard curve derived after 1 h incubation with resorufin standards). To express SSAO activity in nmol H_2_O_2_/h/mg protein, the nmol H_2_O_2_/mL values were divided over the protein concentration (mg/mL), which was previously calculated using the Bicinchoninic acid (BCA) assay.

The data for each methylamine and aminoacetone substrate concentration was transferred to an Excel spreadsheet and analyzed before plotting SSAO activity (nmol H_2_O_2_/h/mg protein) against time (h). The reaction velocity (V) expressed as (nmol H_2_O_2_/h) was derived from the slope of the linear part of the progress curve from the SSAO activity (nmol H_2_O_2_/h/mg protein) vs. time graph for each substrate concentration. SSAO’s kinetics (Km and Vmax) were determined by plotting reaction velocity (nmol H_2_O_2_/h) versus substrate concentration using the non-linear regression model of Michaelis–Menten Y = Vmax ∗ X/(Km + X) on Graph Pad Prism 7 software (version 7.05, San Diego, CA, USA).

### 4.7. Reactive Oxygen Species (ROS) Assay 

Prior to measuring the ROS initial experiment was first performed to establish the optimal time for ROS measurement. Rat aortic vascular smooth muscle cells were plated at 3 × 10^4^ cells/100 μL in a black 96-well plate and allowed to grow for 24–48 h to reach confluence. A ROS red-staining solution was prepared by adding 15 μL of ROS red dye to a 10 mL assay buffer. Confluent cells were washed with 1× PBS and ROS red-staining solution (80 μL) was added to each well before incubation at 37 °C/5% CO_2_ for 1 h. 

Afterward, incubation cells were treated with different SSAO substrates, including benzylamine (500 μM), methylamine (500 μM) and aminoacetone (45 μM), all of which were previously diluted in 1× PBS. 1× PBS (10 μL/well) was used for untreated cells and 5 μM (20 μL/well) 2,3-dimethoxy-1,4-naphthguinone (DMNQ) was used like a positive control. DMNQ is a redox cycling agent that generates both superoxide and hydrogen peroxide intracellularly; it does not react with free thiol groups, is non alkylating and non-adduct forming in contrast to other quinones [53]. To induce ROS production, cells were incubated at 37 °C, and the reading was taken after 15, 30, 60 and 120 min using a Clario Star^®^ Microplate Reader (BMG Labtech) with Ex/Em = 520/605 nm. The plate was kept in incubator at 37 °C/5% CO_2_ between readings.

In the subsequent set of experiments, confluent cells (after previously been incubated for 1 h at 37 °C/5% CO_2_ with ROS red dye) were treated with benzylamine (500 μM), methylamine (500 μM),and aminoacetone (45 μM), with and without the presence of reversible competitive inhibitor of SSAO, β-aminopropionitrile (βAPN) (200 μM), or with and without the presence of mechanism-based, suicide inhibitor of SSAO, MDL72527 (100 μM). 1× PBS (10 μL/well) was used for untreated cells and 5 μM (20 μL/well) of 2,3-dimethoxy-1,4-naphthguinone (DMNQ) was used like a positive control. The cells were incubated for 30 min at 37 °C/5% CO2 and the readings were taken using Clario Star^®^ Microplate Reader (BMG Labtech) with Ex/Em = 520/605 nm.

### 4.8. Measurement of Total Glutathione (GSH)

Total GSH was assessed with a colorimetric recycling assay based on the glutathione recycling system by DTNB (Ellman’s reagent) and glutathione reductase [54]. Rat aortic VSMCs were plated at 5 × 10^5^ cells/1 mL/well in a 24-well plate and allowed to grow for 24–48 h to reach confluence. Confluent cells were treated with benzylamine (500 μM), methylamine (500 μM) and aminoacetone (45 μM), with and without the presence of βAPN (200 μM), or with and without the presence of MDL72527 (100 μM) before incubation at 37 °C/5% CO_2_ for 30 min. Samples were prepared by washing the cells with sterile 1× PBS, scrapping, and centrifugation at 700× *g* for 5 min at 4 °C, after which the pellet was washed with 0.5 mL 1× PBS and centrifuged again at 700× *g* for 5 min at 4 °C. The pellet was then lysed with 80 μL ice-cold glutathione buffer and incubated on ice for 10 min, after which 20 μL of 5% sulfosalicylic acid (SSA) was added, mixed well, and centrifuged again at 8000× *g* for 10 min. The supernatant was transferred to a fresh centrifuge tube and kept on ice ready to be used for the glutathione assay. Reaction mixture was prepared with: NADPH generating mix 20 μL/well, glutathione reductase 20 μL/well and glutathione reaction buffer 120 μL/well.

Furthermore, 1 mL of 1% SSA was added to GSH standard to generate 1 μg/μL glutathione solution, which was then further diluted with 1% SSA to generate 10 ng/μL stock. Next, 10 ng/μL GSH stock was used to prepare GSH standards. Moreover, 160 μL of the reaction mixture was added to each well and the plate was incubated at room temperature for 10 min to generate NADPH. After 10 min of incubation, 20 μL of either GSH standards or samples was added to each well containing reaction mixture and the plate was incubated at room temperature for another 10 min. Next, 20 μL of DTNB was added to each well containing GSH standards and samples and the plate was incubated at room temperature for another 10 min. The absorbance was read on a Clario Star® plate reader (BMG Labtech) and set at 412 nm. The data was transferred to an Excel spreadsheet and analyzed before plotting the absorbance ratio (412 nm) versus concentration of GSH standards (ng/µL). Total GSH was calculated as follows: Total GSH = (Abs sample − Abs blank)/slope STD curve. These values were then corrected for total protein concentration by subtracting them with the protein values obtained from the BCA assay (previously performed) and the total GSH content was expressed as nmol of GSH per mg of total cellular protein.

### 4.9. Statistical Analysis 

The data was analyzed with the statistical software GraphPad Prism 7 (San Diego, CA, USA). Statistical comparisons were made using one or two-way ANOVA, followed by Dunnett’s or Tukey’s multiple comparison tests. Additionally, SSAO kinetics were analyzed with the non-linear regression model of Michaelis-Menten Y = Vmax ∗ X/(Km + X). Probability values < 0.05 were considered as being statistically significant.

## 5. Conclusions

These findings could potentially associate SSAO catalytic activity with the early developing stages of atherosclerosis and vascular damage through induced cellular toxicity, increased ROS levels and a reduction in total GSH. Furthermore, this data shows that methylglyoxal and formaldehyde generate quickly in rat aortic VSMCs because of the SSAO catalyzed reaction. We also noted a higher SSAO affinity for aminoacetone compared to methylamine, which indicates a greater production of methylglyoxal compared to formaldehyde in these cells.

Additional in vitro transcriptional and biochemical studies are needed to fully explore associated signaling pathways related to cellular toxicity and increase in ROS levels. This would provide insight into the potential mechanisms of these transduction pathways involved in the up- and downstream of SSAO’s catalytic activity. Furthermore, considering the different pathways that could lead to reduced cell viability, future in vitro studies are needed to investigate VSMCs phenotype, while focusing on cellular contraction capacity, inflammation and cellular senescence. In addition, in vivo studies using different gender ApoEd/d or LDLR^−/−^ mice fed with a high fat diet and treated with MDL72527 would strongly corroborate the translational significance of the results presented here and shed more light on the sex-related issues of pathogenesis of atherosclerosis.

## Figures and Tables

**Figure 1 ijms-24-04946-f001:**
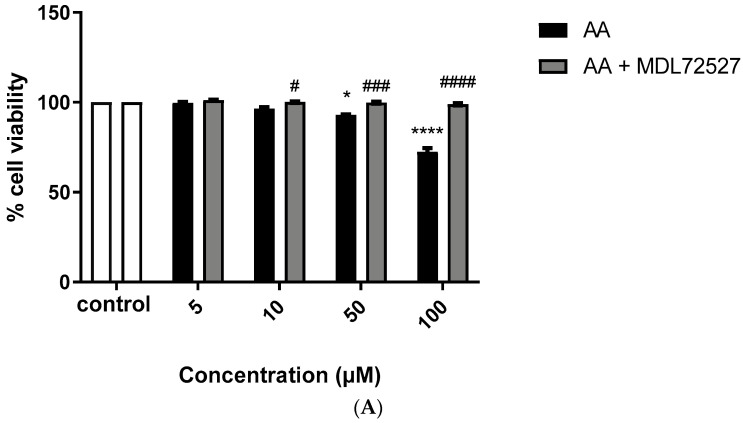
Active SSAO induces VSMCs death, expressed as percentages of cell viability (MTT assay). VSMCs were treated with increasing concentrations of aminoacetone (AA), or AA + 100 µM irreversible SSAO inhibitor MDL72527 (**A**) and increasing concentrations of methylamine (M), or M + 100 µM irreversible SSAO inhibitor MDL72527 (**B**). For figure (**A**), * *p* < 0.05 for 50 µM AA vs. control and **** *p* < 0.0001 for 100 µM AA vs. control (two-way ANOVA followed by Dunnett’s); ^#^
*p* < 0.05 for 10 µM AA + 100 µM MDL72527 vs. AA; ^###^
*p* < 0.001 for 50 µM AA + 100 µM MDL72527 vs. AA; ^####^
*p* < 0.0001 for 100 µM AA + 100 µM MDL72527 vs. AA (two-way ANOVA followed by Tukey’s). For figure (**B**) **** *p* < 0.0001 for 1000 µM M vs. control (two-way ANOVA followed by Dunnett’s); ^####^
*p* < 0.0001 for 1000 µM M + 100 µM MDL72527 vs. M (two-way ANOVA followed by Tukey’s). Values are mean ± S.E.M. (n = 5).

**Figure 2 ijms-24-04946-f002:**
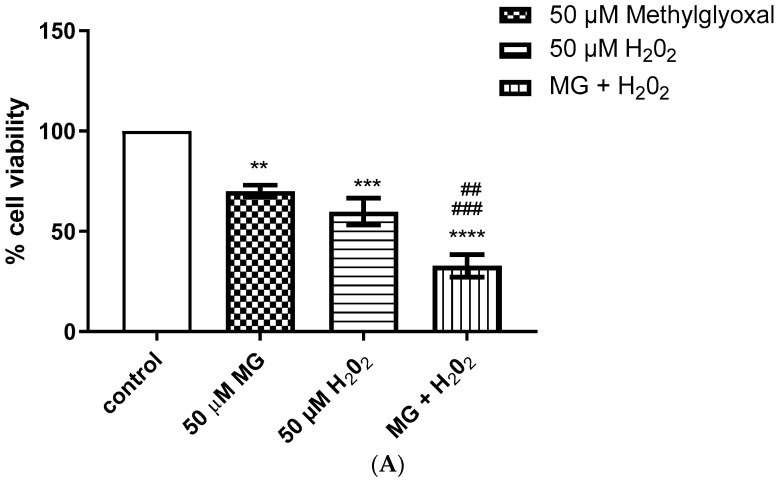
Direct cytotoxic effect was detected from SSAO’s derived products methylglyoxal, formaldehyde and H_2_O_2_, expressed as percentages of cell viability (MTT assay). VSMCs were treated with methylglyoxal (MG), formaldehyde (F), H_2_O_2_, and the combination of both (MG + H_2_O_2_) and (F + H_2_O_2_). For (**A**) ** *p* < 0.01 for 50 µM MG vs. control, *** *p* < 0.001 for 50 µM H_2_O_2_ vs. control, **** *p* < 0.0001 for 50 µM MG + H_2_O_2_ vs. control (one-way ANOVA followed by Dunnett’s), ^###^
*p* < 0.001 for 50 µM MG + H_2_O_2_ vs. 50 µM MG; ^##^
*p* < 0.01 for 50 µM MG + H_2_O_2_ vs. 50 µM H_2_O_2_ (one-way ANOVA followed by Tukey’s). For (**B**) ** *p* < 0.01 for 1000 µM F vs. control, **** *p* < 0.0001 for 1000 µM H_2_O_2_ vs. control and **** *p* < 0.0001 for 1000 µM F + H_2_O_2_ vs. control (one-way ANOVA followed by Dunnett’s); ^##^
*p* < 0.01 for 1000 µM H_2_O_2_ vs. 1000 µM F; ^###^
*p* < 0.001 for 1000 µM F + H_2_O_2_ vs. 1000 µM F (one-way ANOVA followed by Tukey’s). Values are mean ± S.E.M. (n = 5).

**Figure 3 ijms-24-04946-f003:**
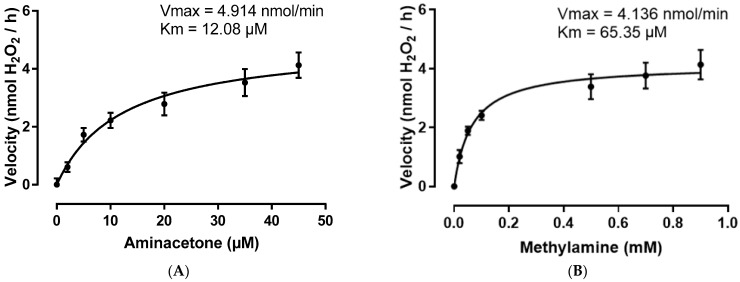
SSAO kinetic parameters (Vmax and Km) in the presence of aminoacetone (**A**) and methylamine (**B**) as substrates. The data was analyzed with the non-linear regression model of Michaelis-Menten Y = Vmax ∗ X/(Km + X). Values are mean ± S.E.M. (n = 5).

**Figure 4 ijms-24-04946-f004:**
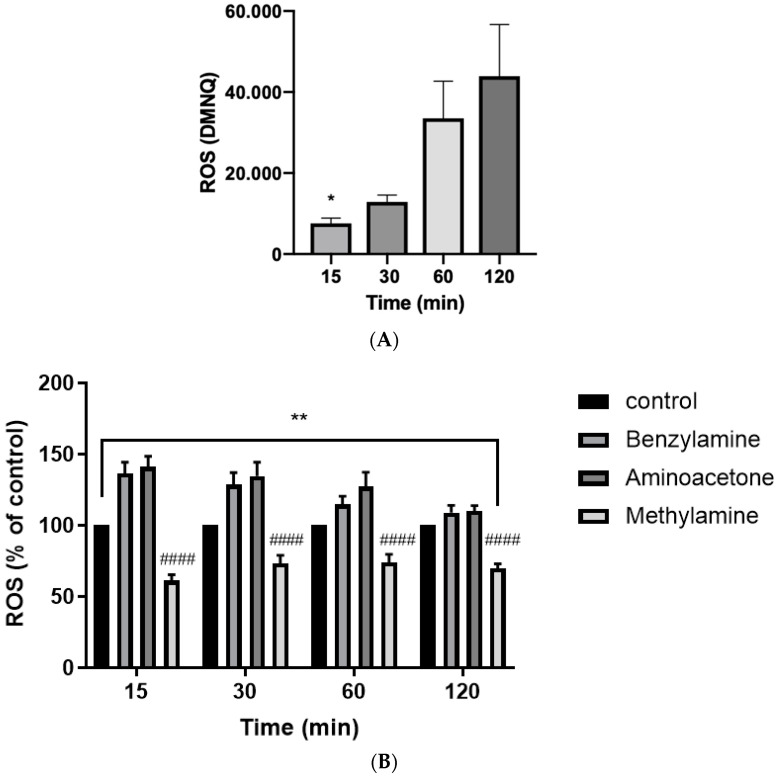
SSAO activity induces ROS formation in rat aortic VSMCs. In (**A**), VSMCs were treated with 5 μM (20 μL/well) DMNQ and ROS was measured over different time intervals. Significant difference in ROS production was detected over time (* *p* < 0.05 for 15 min vs. 120 min, one-way ANOVA followed by Tukey’s). In (**B**), ROS production (expressed as percentage of control) was compared over time (*) and after different amine treatments (#), benzylamine (500 μM), aminoacetone (45 μM) and methylamine (500 μM). Significant difference in ROS formation was detected over time (** *p* < 0.01 for 15 min vs. 120 min, *p* < 0.01 for 30 min vs. 120 min, *p* > 0.05 for 60 min vs. 120 min, two-way ANOVA followed by Tukey’s). Multiple comparison (Tukey’s) then compared between different amine treatments. At 15 min incubation, ^####^
*p* < 0.0001 for methylamine vs. benzylamine and ^####^
*p* < 0.0001 for methylamine vs. aminoacetone. At 30-, 60-, and 120-min incubation ^####^
*p* < 0.0001 for methylamine vs. benzylamine and ^####^
*p* < 0.0001 for methylamine vs. aminoacetone The asterisk (*) indicates statistical significance in ROS production detected over time. The hash (^#^) indicates statistical difference in ROS production detected between treatments at each respective time point. Values are mean ± S.E.M. (n = 5).

**Figure 5 ijms-24-04946-f005:**
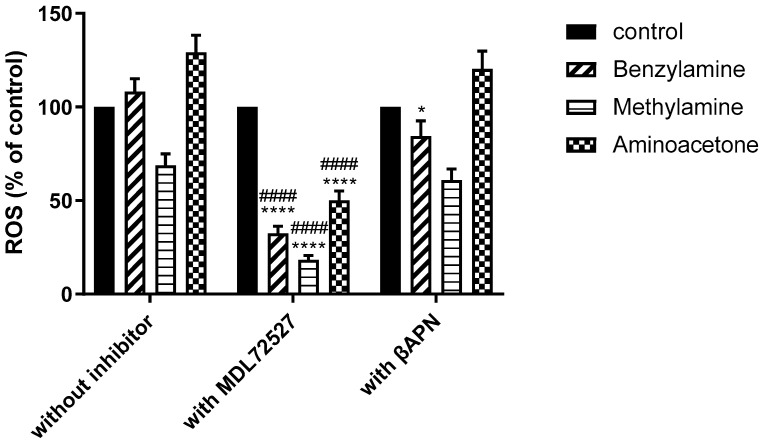
MDL72527 demonstrates stronger inhibitory potential over βAPN in inhibiting SSAO driven ROS production in VSMCs. VSMCs were treated with different amines, benzylamine (500 μM), aminoacetone (45 μM) and methylamine (500 μM), and amines in the presence of inhibitors (100 μM MDL72527) and (200 μM βAPN). In benzylamine-treated cells there was a significant difference in ROS production in cells without inhibitor and in the presence of MDL72527 and βAPN (**** *p* < 0.0001 for MDL72527 vs. without inhibitor & * *p* < 0.05 for βAPN vs. without inhibitor, two-way ANOVA followed by Tukey’s). In methylamine- and aminoacetone-treated cells there was a significant difference in ROS production in cells without inhibitor and in the presence of MDL72527 (**** *p* < 0.0001 for MDL72527 vs. without inhibitor, two-way ANOVA followed by Tukey’s). There was no statistical difference (*p* > 0.05) in ROS production in methylamine- and aminoacetone-treated cells without inhibitor and in the presence of βAPN. Additionally, significant difference in ROS production was detected between MDL72527 and βAPN in benzylamine, methylamine- and aminoacetone-treated cells (^####^
*p* < 0.0001 for MDL72527 vs. βAPN, two-way ANOVA followed by Tukey’s). The asterisk (*) indicates statistical difference between cells without inhibitor and cells treated with MDL72527 or βAPN. The hash (^#^) indicates statistical difference between MDL72527 and βAPN for each amine. Values are mean ± S.E.M. (n = 5).

**Figure 6 ijms-24-04946-f006:**
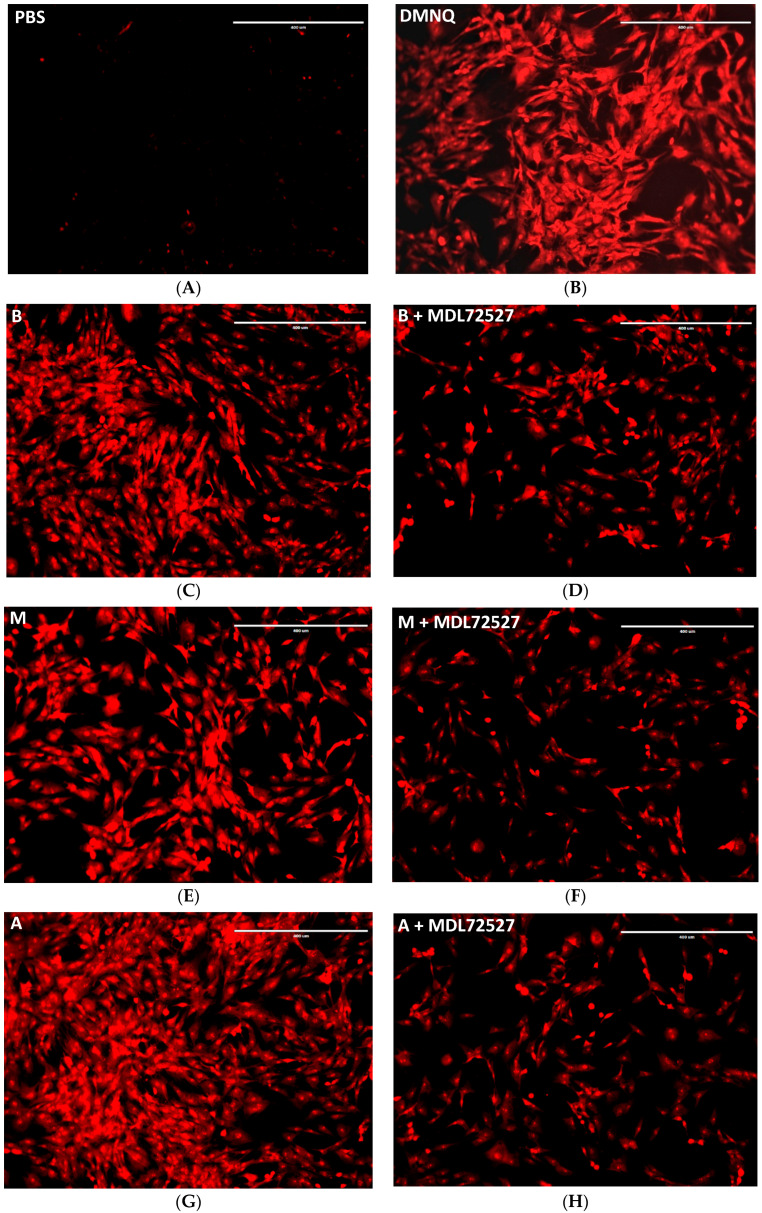
Fluorescent images depicting ROS production in: cells treated with 1× PBS ((**A**), negative control), 5 μM DMNQ ((**B**), positive control), 500 μM benzylamine (**C**), 500 μM benzylamine + 100 μM MDL72527 (**D**), 500 μM methylamine (**E**), 500 μM methylamine + 100 μM MDL72527 (**F**), 45 μM aminoacetone (**G**), 45 μM aminoacetone + 100 μM MDL72527 (**H**). Scale bar: 400 μM, magnification is 10×.

**Figure 7 ijms-24-04946-f007:**
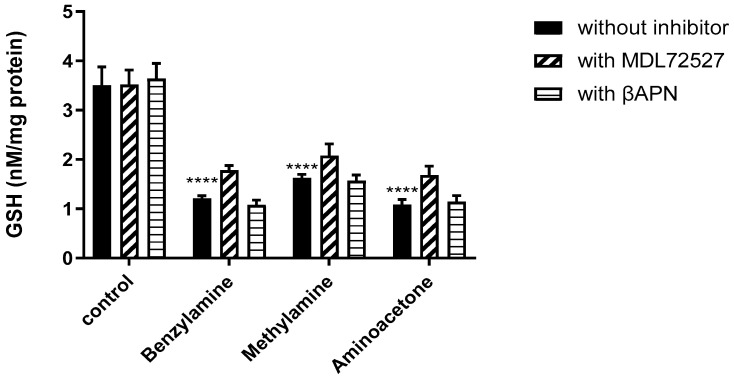
SSAO activity reduces total GSH (nM/mg protein) in rat aortic VSMCs. VSMCs were treated with benzylamine (500 μM), methylamine (500 μM), and aminoacetone (45 μM) (without inhibitor), and the amines in the presence of 100 μM MDL72527 (with MDL72527), and 200 μM βAPN (with βAPN). For cells without the inhibitor (**** *p* < 0.0001 for benzylamine, methylamine and aminoacetone treated cells vs control, two-way ANOVA followed by Dunnett’s); cells treated with MDL72527 (**** *p* < 0.0001 for benzylamine, methylamine, and aminoacetone vs control, two-way ANOVA followed by Dunnett’s); and cells treated with βAPN (**** *p* < 0.0001 for benzylamine, methylamine and aminoacetone treated cells vs control, two-way ANOVA followed by Dunnett’s). In benzylamine, methylamine and aminoacetone treated cells there was no statistical difference between cells without inhibitor and cells treated with MDL72527 and βAPN (*p* > 0.05). The asterisk (*) indicates statistical significance between benzylamine, methylamine and aminoacetone treated cells vs control. Values are mean ± S.E.M. (n = 5).

## Data Availability

The data presented in this study are available on request from the corresponding author. The data are not publicly available due to privacy concerns.

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
