# Peer review of "The Impact of Semicarbazide Sensitive Amine Oxidase Activity on Rat Aortic Vascular Smooth Muscle Cells"

_ijms, 2023, doi:10.3390/ijms24054946_

Round 1

Reviewer 1 Report

Methylamines are important risk factors for atherosclerosis. It has been demonstrated that some compounds present in olive oil can inhibit production of TMA in rat intestine and thus decrease development of atheroma (https://www.cell.com/cell/fulltext/S0092-8674(15)01574-3). It has also been shown in 1993 that methylamine does not appear to harm endothelial cells at concentrations of up to 100mmol/L [i.e. 100,000 umol/L to emphasise that a very high concentration can be tolerated] unless SSAO is added (in serum) to the test platform ( https://go.gale.com/ps/i.do?p=AONE&u=googlescholar&id=GALE|A13715387&v=2.1&it=r&sid=googleScholar&asid=bdc36fcc ) and that the effect is due to formaldehyde toxicity. Surprisingly, none of this early work was referenced in this paper. Apart from this minor point, the introduction covers the science well.

The science in this paper looks at measures of cell death which was shown to be reversible with the inhibitor MDL72527; A similar effect was shown in 1993, except MDL72974A was used. Similarly, the 1993 paper also assessed the effect of hydrogen peroxide and formaldehyde. Some new assays were used, such as Amplex red assay, ROS species assay and GSH assay, so there is new information in this paper but as there is a publication from 30 years ago that has a substantial amount of the information in this paper, one would expect that it would at least be referenced.

Author Response

Dear Reviewer, 

Thank you for taking the time to read the manuscript and provide suggestions for improvement. 

We have updated the manuscript as per your suggestions. The updates have been highlighted in red. 

Kind regards,

Vesna

Reviewer 2 Report

The article is devoted to the study of the impact of semicarbazide sensitive amine oxidase (SSAO) activity on rat aortic vascular smooth muscle cells. 

The introductory section contains the information on the connection of SSAO with atherosclerosis and other cardiovascular diseases. However it would be nice to provide a clear definition of atherosclerosis as well as to mention pathological factors contributing or potentially contributing to its development (such as multiply modified low-density lipoproteins, including desialylated and oxidized ones, mutations of mitochondrial DNA and mitochondrial dysfunction). Also, there are differences in atherosclerosis development between genders. Thus, it is important to mention the complexity of atherosclerosis pathogenesis since up to now there is no clear understanding of the causes of atherosclerosis which is connected to the lack of effective medication and approaches for anti-atherosclerotic therapy (statin therapy is not effective in 100% cases). Some insights for the discussion can be found for example here: DOI10.3389/fcvm.2021.707889 ; 10.3390/ijms22084080 .

Also, the authors should explain why they chose rat model for experiments since it is not a golden standard for research in atherosclerosis field (rats do not develop atherosclerosis). Even apoE(-/-) mice arguably is not the best choice to translate the results of research on humans, but it is widely used.

Overall, the experiments done are well-described and conclusions are correct. However, there are some issues to be addressed.

Figure 1 legend as well as the line 270 should both include the mention that MDL72527 is SSAO inhibitor. Figure 1 and 2legends should also mention the method of detection of cell survival usedThe source of MDL72527 should be mentioned in Methods section.

Cytotoxic concentrations of aminoacetone, and methylamine were determined for rat aortic VSMCs. Enhanced cytotoxic effect of SSAO-derived products was observed after simultaneous addition of methylglyoxal and hydrogen peroxide, and formaldehyde and hydrogen peroxide. The authors should discuss the correspondence of aminoacetone, methylamine,methylglyoxal, formaldehyde, and hydrogen peroxide concentrations studied with physiologically achievable concentrations of these substances. 

Line 320. Km value for methylamine differs from the one mentioned in Figure 3 (μM vs mM). The manufacturer of SSAO used in this experiment should be provided in Materials section. Where does this enzyme originate from? Does it have a sequence of human SSAO or rat SSAO etc? If cell lines were used for these experiment then the question appears whether there are other than SSAO enzymes potentially contributing to enzymatic activities studied. The way of measuring of enzyme kinetics should be described in Methods section.

Line 330. There is a typo in word “SSAO”.

Lines 347-349. “Significant difference in ROS formation was detected over time (**p < 0.01 for 15 min vs 120 min and **p < 0.01 for 30 min vs 120 min, two-way ANOVA followed by Tukey’s) and between different amine treatments.” This statement is a little bit confusing since Figure 4 shows two cases labelled with two asterisks (**), and both of these cases are showing differences inside 15- and 30-minutes time groups between control and methylamine. Also, the statement cited does not seem to be truth at least for methylamine.

What were the concentrations used of benzylamine, methylamine, aminoacetone in Figures 4 and 5? It should be shown in figure legends.

Lines 420. Reference/s should be provided.

Lines 501-502. “Our data shows highest ROS production after aminoacetone treatment, followed by benzylamine and then methylamine (Figure 4B & Figure 5).” It would be nice to know the concentrations of substances used, and whether these concentrations were identical.

Lines 555-556. “These findings associate SSAO catalytic activity with the early developing stages of atherosclerosis…” I would suggest to “soften” this statement by adding the word “potentially” because this research is not directly studying changes happening during atherosclerosis. The cells used for the research are not purified from initial atherosclerotic lesions, rats normally do not develop atherosclerosis etc. But the results obtained are still important and potentially can contribute to atherosclerosis field by providing important insights. 

List of abbreviations used would benefit the article.

The paper can be accepted after minor revision.

Author Response

(The authors gave the same response as above.)

Round 2

Reviewer 1 Report

Previous work now cited. Other changes acceptable.